# PD-1/PD-L1, MDSC Pathways, and Checkpoint Inhibitor Therapy in *Ph*(-) Myeloproliferative Neoplasm: A Review

**DOI:** 10.3390/ijms23105837

**Published:** 2022-05-23

**Authors:** Jen-Chin Wang, Lishi Sun

**Affiliations:** Division of Hematology/Oncology, Brookdale University Hospital Medical Center, Brooklyn, NY 11212, USA; lishi.sun@gmail.com

**Keywords:** myeloproliferative neoplasm, myeloid-derived suppressor cells, PD-1, PD-L1, myeloid suppressor cells (MDSC), immune checkpoint inhibitor therapy (CPI)

## Abstract

There has been significant progress in immune checkpoint inhibitor (CPI) therapy in many solid tumor types. However, only a single failed study has been published in treating *Ph(*-) myeloproliferative neoplasm (MPN). To make progress in CPI studies on this disease, herein, we review and summarize the mechanisms of activation of the PD-L1 promoter, which are as follows: (a) the extrinsic mechanism, which is activated by interferon gamma (IFN γ) by tumor infiltration lymphocytes (TIL) and NK cells; (b) the intrinsic mechanism of EGFR or PTEN loss resulting in the activation of the MAPK and AKT pathways and then stat 1 and 3 activation; and (c) 9p24 amplicon amplification, resulting in PD-L1 and Jak2 activation. We also review the literature and postulate that many of the failures of CPI therapy in MPN are likely due to excessive MDSC activities. We list all of the anti-MDSC agents, especially those with ruxolitinib, IMID compounds, and BTK inhibitors, which may be combined with CPI therapy in the future as part of clinical trials applying CPI therapy to *Ph*(-) MPN.

## 1. Introduction

In 2018, James P Allison and Tasuku Honjo were awarded the Nobel Prize in Physiology or Medicine. They revolutionized cancer therapy with immune checkpoint blockade therapy [1]. Dr. Allison showed that the CTLA-4 monoclonal antibody blocked CTLA-4 function, which enhanced anti-tumor immunity and led to a cure for cancer in mouse models [2]. This monoclonal antibody to human CTLA-4 was approved by the Food and Drug Administration (FDA) in 2011 to treat malignant melanoma [3]; this was the first clinical use of an immune checkpoint inhibitor (CPI) in cancer treatment. Dr. Honjo was the first to clone PD-1, in 1992 [4], and he described PD-1 as another important immune checkpoint of T cells [5]. Subsequently, his work led to identifying its ligands as B7-H1 [6] or PD-L1 [7]. Subsequent extensive studies on the PD-1/PD-L1 pathway led to multiple FDA-approved blocking antibodies against PD-1 or PD-L1 for the treatment of a broad spectrum of human cancers.

Over the years, clinical trials involving these two checkpoint inhibitors (anti-PD-1/PD- L1 and CTLA-4) and CPI therapy as novel anticancer therapeutics have been undertaken, inducing lasting tumor regression in more than 14 malignancies, including metastatic lung cancer, melanoma, Hodgkin’s lymphoma, colon carcinoma, urothelial carcinoma, and many others. Three PD-1 inhibitors (nivolumab, pembrolizumab, and cemiplimab) and three PD-L1 inhibitors (atezolizumab, avelumab, and durvalumab) are now approved in addition to ipilimumab for CPI therapy [8,9].

Although there has been much progress in the use of CPI therapy to treat cancer, very few studies have been published regarding the CPI pathways in *Ph*(-) myeloproliferative neoplasm (MPN). At the American Society of Hematology (ASH) annual meeting in 2020, Mascarenhas and his associates [10] presented a multi-center, open-label, phase 2, single-arm study of pembrolizumab in patients with primary, post-essential thrombocythemia or post-polycythemia vera myelofibrosis (MF) (NCT03065400). Nine case studies were presented, but none had a clinical response. We published an article demonstrating that PD-1 and PD-L1 were increased in MPN disease in immune cells, including CD4, CD8, monocyte, and CD34+ cells [11]. Prestipino et al. showed that oncogenic Janus kinase 2 (JAK2) activity caused the signal transducer and activator of transcription 3 (stat 3) and stat 5 phosphorylation, which enhanced PD-L1 promoter activity and PD-L1 protein expression in JAK2V617F-mutant cells [12]. 

Therefore, only a few studies have been published regarding CPIs in MPN. Herein, we review the PD-L1 activation mechanism and MDSCs’ interactions with PD-1 and PD-L1, and we propose possible avenues for advancing CPI therapy in MPN. 

## 2. Mechanisms of Increased PD-L1 Expression 

The PD-L1 promoter gene sequence and transcription factor activation site have been described (Figure 1) [13,14,15,16]. However, a comprehensive study of the mechanism of PD-L1 expression in tumor cells has not been undertaken. Based on our literature review, three major mechanisms may regulate the PD-L1 gene. First is an “extrinsic” mechanism where an anti-tumor cellular immune response is driven by micro-environment cells, such as natural killer cells (NK) and CD8+ tumor-infiltrating lymphocytes (TILs), producing IFNγ; this, in turn, may induce PD-L1 expression on tumor cells. Second, an “intrinsic” mechanism may exist in which constitutive oncogenic signaling pathways within the tumor cell lead to PD-L1 overexpression. The third mechanism is 9p24.1 gene amplification. The PD-1 ligand genes, PD-L1 and PD-L2, are located on chromosome 9p24.1 and separated by only 42 kilobases. In addition, 9p copy gain has been described in Hodgkin’s lymphoma (HL) and primary mediastinal large B-cell lymphoma (MLBCL), leading to increased PD-L1 amplification and identifying the PD-1 ligands as key targets of the 9p24.1 amplification in NSHL and MLBCL.

### 2.1. The Extrinsic (IFNγ) Pathway

Interferon (IFN) was first described in the 1950s as an agent from T cells or NK cells that interferes with viral replication [16]. Signaling from the interferon receptors was subsequently well-characterized [17,18,19]. Janus kinase (JAK) and the signal transducer and activators of transcription (STAT) were found to be the main signaling pathways mediating interferon-induced gene expression [19,20] and resulting in the activation of interferon-stimulated response elements (ISREs) [19,20] and gamma interferon activation sites (GASs) [21,22].

IFN-γ plays key signaling roles through the interferon-gamma–JAK1/JAK2–STAT1/STAT22/STAT33–IRF1 axis (resulting in the binding of the IRF1 transcription factor to the PD-L1 promoter (ISRE_s_ and GAS_S_)) and a weaker binding to the PD-L2 promoter (inducing PD-L1, L2 formations) [23,24]. Mutations of this interferon signaling pathway, including Jak1, 2, were also associated with primary and acquired resistance to immune checkpoint blockade therapy for cancer [14,25,26]. 

Tumor-infiltrating lymphocytes (TILs) have long been associated with the clinical treatment response in various solid cancers [27,28], and the tumor microenvironment has long been recognized as an important prognostic factor. The expression of PD-L1 and PD-1 in early breast cancer is associated with higher TIL scores and clinical–pathological response. PD-L1 expression in tumor cells was significantly associated with PD-1 expression in TILs (*p* = 0.03) in breast carcinoma [29]. In other tumor types, such as head and neck carcinoma, in the tumor microenvironment (TME), T regulator cells (Tregs) are highly activated, but conventional T cells (Tconvs) are exhausted or inactivated by Tregs, and Tregs are strongly involved in the creation of an immunosuppressive environment in HNSCC tissues. Thus, Treg-targeting drugs are expected to be a combination partner with immune-checkpoint inhibitors that will improve the immunotherapy of HNSCC [30]. In non-small cell lung carcinoma, the number of TILs and PD-1 expression on TILs in B7-H1-positive tumor regions was found to be significantly lower than those in negative tumor regions. These results suggest that the expression of B7-H1 on tumor cells might contribute to negative regulatory immune responses against TILs in non-small cell lung cancer [31,32]. Therefore, TILs in relation to PD-1 and PD-L1 expression on the tumor cells may be different in different types of tumors. 

### 2.2. The Intrinsic Mechanism (Oncogene Growth Factor Pathway)

The constitutively increased expression of PD-L1 associated with the oncogenic growth factor can be accounted for in two ways. First, (IIA) the PD-L1 protein may be up-regulated after EGFR/RAS/MAPK pathway activation, either by over-expression of wild types of EGFR, such as in head and neck carcinoma [16], or by activating mutations of RAS and EGFR, which is more seen in non-small cell lung carcinoma (NSCLC) [33,34,35,36]. Wild-type EGFR activates PD-L1 through the JAK2/STAT1 pathway [16] (Figure 1). In contrast, mutant EGFR/RAS may more strongly induce MAPK pathway activation [13,35,36]. MAPK signaling increases STAT signaling, thus adding to the transcriptional activity. MAPK signaling also increases the stability of CD274 mRNA. The RAS/EGFR mutant induces CD274 expression or PD-L1 mostly by the signal transmitter and transcription factor STAT3, which then binds to the CD274 gene promoter, as shown in vitro and in vivo [37].

Second, the (IIB) functional loss of tumor-suppressive genes such as phosphatase and tensin homolog (PTEN) in glioma, and the loss of Lkb1 (also known as serine-threonine kinase 11 [Stk11]) and PTEN in squamous cell carcinoma of the lung, results in the activation of the phosphatidylinositol-3-OH kinase (PI3Kinase) pathway, then the activation of the AKT pathway, followed by the PD-L1 promoter; thus, the expression of the PD-L1 protein seems to be dependent on Akt activation in some cases of glioma [38] and lung cancer [39]. 

### 2.3. 9p24.1 Gene Amplicon Amplification

The PD-1 ligand genes, PD-L1 and PD-L2, are located on chromosome 9p24.1 and separated by only 42 kilobases. A 9p copy gain has been described in both Hodgkin’s lymphoma and primary mediastinal large B-cell lymphoma (MLBCL) [15]. By laser-capture microdissection and quantitative immunohistochemical methods in primary Hodgkin Reed–Sternberg cells and primary MLBCLs, it was found that programmed cell death 1 (PD-1) ligand/9p24.1 amplification was found in nodular sclerosing HL and MLBCL, with a correlation between 9p24.1 copy number and PD-1 ligand expression in primary tumors. The amplification region also included the Janus kinase 2 (*JAK2*) locus. The Jak2 amplification also activates stat 1 and induces PD-1/PD-L1 transcription [12].

In addition, 9p24.1 gene amplicon amplification has been described not only in lymphoma but also in other types of neoplasms. In carcinoma of the lung, in one study, it was found that 5 of 89 samples had PD-L1 gene amplification; the patients with PD-L1 amplification had worse prognoses than those without. This genetic amplification of the PD-L1 gene was also correlated with JAK2 gene amplification [40]. It was additionally observed in 32 of 48 cervical squamous cell carcinomas (SCC) (67%) and 10 of 23 vulvar SCCs (43%) [41]. A high level of 9p24 amplicon (log_2_ ratio of ≥1) was also found in triple-negative breast carcinoma (TNBC) (12 of 41 cases), glioblastomas (2 of 44 cases), and colon carcinomas (2 of 68 cases) [42]. 

## 3. PD-1 and PD-L1 Interaction with MDSCs

### 3.1. MDSC (Myeloid Suppressor Cell) Function

(1)The expression of NOX2, iNOS, and Arginase 1. In contrast to T cells, which present immune-checkpoint molecules on their surface (including cytotoxic T-lymphocyte-associated antigen 4 (CTLA-4), lymphocyte-activation gene 3 (LAG-3), programmed cell death protein 1 (PD-1), T-cell immunoglobulin and mucin-domain containing 3 (TIM-3), and B and T lymphocyte attenuator (BTLA)) [43], MDSCs mainly exert their mechanism as an immune modulator through inducing the expression of NADPH oxidase (NOX2) and inducible nitric oxide synthase (iNOS), which generates different ROS compounds and nitric oxide (NO). It is known that the product of H_2_O_2_ and NO, i.e., peroxynitrite (ONOO−), can nitrate the 22 tyrosine residues of the T-cell receptor (TCR), after which, the receptor no longer recognizes antigen peptides; thus, the TCR signaling pathway is inhibited [44,45]. MDSCs also express arginase I [46], which inhibits and possibly deletes tumor-specific cytotoxic T cells (CTL): CD3ς is lost in T cells exposed to arginase I [47,48]. Overall, granulocyte MDSCs (G-MDSCs) have higher arginase-1, MPO, and ROS activities [49], and monocytic MDSCs (M-MDSCs) are mediated by arginase-1, NO, and different soluble factors [50]. Several other suppressive mechanisms from MDSCs have also been suggested: the secretion of TGF-ß [51,52], induction of regulatory T cells [53,54,55], depletion of cysteine [56], and up-regulation of cyclo-oxygenase 2 (cox2) and prostaglandin E2 (PGE2) [47].(2)MDSCs can show an up-regulation of PD-L1 [57] and CD 80 [58] on their surface to exert immunosuppression. PD-L1 is activated through IFN_Γ_-stat 1 activation [59,60,61,62]. IFN_Γ_ is highly expressed in cells of the tumor tissues, and, through phosphorated STAT 1, it binds to a unique IRF-binding sequence element in vitro and chromatin in vivo in the cd274 promoter to activate PD-L1 transcription. In addition to IFN_Γ_, IL-10, VEGF, and hypoxia are other novel critical modulators of PD-L1 expression in MDSCs [63,64]. The expression of CD80, normally found on dendritic cells or macrophages, was increased in MDSCs in patients with malignant melanoma, and it was also up-regulated on MDSCs in a murine ovarian cancer model; its ligation of CTLA-4 through CD80 on Tregs is crucial for T cell suppression [54]. Treg cells are also able to stimulate B7-H1 expression in myeloid-derived suppressor cells [65], so MDSCs and Tregs co-operatively enhance each other’s immune-suppression functions.(3)Up-regulated myeloid cell receptor tyrosine kinases (RTKs) TYRO3, AXL, and MERTK, and their ligands Gas 6 and Protein S. These RTKs are the physiological pathways used to suppress innate immune responses, including in the tumor microenvironment. Myeloid-derived suppressor cells (MDSCs) can up-regulate TYRO3, AXL, and MERTK and their ligands to exert immunosuppressive functions [66].(4)Accumulation and expansion of MDSCs. These processes are controlled by a network of transcription factors and regulators to expand immature myeloid cells and ensure the pathologic activation of these immature cells. These expansions and activations are indispensable for MDSC accumulation [67]. STAT3, IRF8, C/EBPβ, RB1, and adenosine receptors A2b NLRP3 are the important transcription factors for MDSC expansion, and NF-kß, the STAT1 pathway, the STAT 6 pathway, PGE2 and COX 2, and ER stress are the transcriptional factors for MDSC activation. STAT3 was the first transcription factor implicated in MDSC expansion [67]. S100A9 and S100A8 also expand and activate MDSCs, and forced S100A9 can induce MDSCs and thereby induce clinical myelodysplastic syndrome [67,68,69,70].

### 3.2. MDSC and PD-1 Inhibitor Therapy 

Recent studies suggest a leading role for MDSCs in the immunosuppressive TME, which could be the main cause of therapeutic resistance to CPI therapy [71,72,73]. To summarize, in malignant melanoma, patients who respond to ipilimumab have significantly lower M-MDSC percentages in their peripheral blood, and abnormal MDSC accumulation in patients with advanced melanoma is strongly associated with resistance to immune modification agents [74,75,76,77], as a higher M-MDSC frequency is associated with a decreased expansion and activation of tumor-specific T cells [78].

A strong positive correlation between the MDSC percentage and neutrophil/lymphocyte rate (NLR) (a prognostic marker in both ipilimumab and nivolumab therapy) has also been investigated in patients with breast cancer and non-small cell lung carcinoma [78,79] following PD-1 therapy, showing a correlation of the clinical response with decreased MDSCs or a decreased neutrophil/lymphocyte ratio. In another example, the impact of circulating Tregs, G-MDSCs, and M-MDSCs on anti-PD-1 therapy was assessed in non-small cell lung cancers by flow cytometry [80]. Anti-PD-1 treatment boosted circulating Treg levels; in contrast, anti-PD-1 therapy did not change G-MDSC and M-MDSC levels overall. However, the partial response (PR) group had a higher baseline level of M-MDSCs, which exhibited a significant decrease after the first cycle of anti-PD-1 treatment. Therefore, MDSC accumulation derives a more potent immunosuppressive network within the tumor microenvironment than other immune-suppressive agents, such as Tregs, and MDSCs have an important role in the failure of CPI therapy; CPI therapy may be enhanced by combining it with therapies targeting MDSCs, which may break the CPI therapy resistance. 

### 3.3. Anti-MDSC Therapy 

Anti-MDSC or MDSC-eliminating regimens are important for CPI therapy. The original descriptions by Law et al. [81] are summarized and modified in Table 1. 

(1)Depleting MDSCs

Low-dose chemotherapy, such as 5-fluorouracil (5FU) and gemcitabine, enhances anti-tumor immune activity by depleting MDSCs in tumor-bearing mice [82,83,84,85]. In the TME, VEGF promotes the expansion of MDSCs by recruiting Tregs, inducing angiogenesis, and facilitating tumor progression. Therefore, the tyrosine kinase inhibitor sunitinib has been used successfully in the treatment of patients with renal cell carcinoma by depleting MDSCs via the blockade of VEGF and c-KIT signaling [86,87,88]. In addition, sunitinib was found to inhibit stat 3, which is related to MDSC expansion, so renal cell carcinoma patients treated with sunitinib showed a decrease in MDSC accumulation and, consequently, enhanced T cell function [87]. Ibrutinib treatment also resulted in a significant reduction in MDSCs (expressed BTK) in wild-type mice bearing B16F10 melanoma tumors, but not in X-linked immunodeficiency mice (XID) harboring a Bruton tyrosine kinase (BTK) mutation [89], suggesting that ibrutinib may be a potential therapeutic agent to reduce MDSCs in cancer types other than chronic lymphatic leukemia (CLL).

(2)Blockade of MDSC Migration

(a) The chemokine receptor CCR5, via the ligands CCL3, CCL4, and CCL5, plays a crucial role in the chemotaxis of MDSC into the TME [72]. Thus, the blockade of CCR5 inhibits the recruitment and immunosuppressive activity of MDSCs, and an improved survival in melanoma was demonstrated by Blattner et al. [90]. Similarly, CCR5 antagonists also inhibited the metastatic potential of basal breast cancer and reduced tumor growth [91,92]. (b) CSF-1R: CSF-1R is a tyrosine kinase receptor that, when bound with its ligand CSF-1, not only promotes the migration of MDSCs but also can differentiate and expand myeloid cells into MDSCs and tumor-associated macrophages (TAMs) [93]. CSF-1R is up-regulated in several cancer types, such as pancreatic and breast cancers [94,95]. Treatment targeting the receptor or its ligand, CSF-1R/CSF-1, has been found to improve T cell responses; further, combining CSF-1R inhibition with checkpoint blockades or adoptive T cell transfer therapy resulted in an improved anti-tumor T cell activity and tumor regression [93,96,97]. (c) The NLRP3 pathway: the programmed death-ligand 1/NOD-, LRR-, and pyrin domain-containing protein 3 (PD-L1/NLRP3) inflammasome signaling cascade [97], in cooperation with CXCR2, leads to the recruitment of granulocytic myeloid-derived suppressor cells (PMN-MDSCs) into tumor tissues, thereby blunting the T cell response. Blockage of this NLRP3 pathway leads to a decrease in G-MDSC accumulation and a revitalized T cell response, which could be a promising target for future translational research.

(3)Attenuating MDSC Immunosuppressive Functions

PGE2, by up-regulating arginase-1 in MDSCs, was implicated in the suppression of T cells [47]. PGE2 also increased the expression of COX2 in monocytes, converting them to M-MDSCs; further, MDSCs increase the production of PGE2 [98,99]. PGE2′s MDSC-suppressive function was confirmed in melanoma patients, where the inhibition of COX2 was found to downregulate the suppressive activity of M-MDSCs [100,101]. In two mouse models of glioma and mesothelioma [102,103], the COX2 inhibitor celecoxib blocked PGE2, then blocked MDSC accumulation and arginase production. Therefore, the disruption of COX-2/PGE2 signaling successfully repressed MDSCs, subsequently improving the CTL frequency and immune response, delaying tumor growth, and working in synergy with checkpoint inhibitors [103,104,105]. Anti-inflammatory triterpenoids have also been used to activate the Nrf2 gene in MDSCs. Nrf2 is involved in conferring cytoprotection against oxidative stress [106]. The activation of Nrf2 using synthetic triterpenoids, such as CCDO-IM and CCDO-Me, resulted in a reduced intracellular ROS production and has shown promising anticancer results in Phase 1 clinical trials, with the therapy being well-tolerated by patients [107,108,109].

(4)Inducing MDSC Differentiation

ATRA, by directly inducing the differentiation of MDSCs into mature antigen-presenting precursor cells, resulted in a reduction in T-cell suppression [110]. This reduction in MDSCs and improvement in the T-cell response have been observed both in mice and in patients with various cancer types, such as renal cell carcinoma [111,112]. Chemotherapies were also reported to induce MDSC differentiation into tumor-associated macrophages (TAMs); these effects were observed using the DNA demethylating epigenetic agents 5-azacytidine and docetaxel [113,114]. Ibrutinib was also shown to be able to differentiate MDSCs into dendric cells: tumors from ibrutinib-treated mice contained more mature DCs and lower numbers of myeloid-derived suppressor cells (MDSCs), and the ex vivo treatment of MDSCs with ibrutinib was also found to switch their MDSC phenotype to a mature DC phenotype with a significantly enhanced MHCII expression, subsequently promoting T cell proliferation and effector functions and the induction of anti-tumor TH1 and CTL immune responses [115].

(5)Inhibition of PD-L1 and VISTA Expression on MDSCs

Several mechanisms have been described for the increased PD-L1 expression on MDSCs. We describe four of these mechanisms below. (1) PD-L1 is highly expressed in tumor-infiltrating MDSCs through highly expressed IFN_Γ_ in the tumor tissue, with the pSTAT1–IRF1 axis activating the PD-L1 promoter [57]. (2) Hypoxia and chronic inflammation, which activate NF-KB, can also cause an increased expression of PD-1 and PD-L1 in MDSCs [63,116]. (3) S-100 A9 has also been implicated in the increased expression of PD-1 and PD-L1 in MDSCs in CLL [69], and, when CLL was treated with hypomethylating agents, an increased expression of checkpoint proteins (PD-L1) in MDSCs was reported [117,118], with resistance attributed to a hypomethylating-resistant mechanism. However, the role of PD-L1 in MDSCs in T cell suppression needs further study. (4) Treg cells stimulate PD-L1 expression on MDSCs [65]; Treg cells were shown to stimulate PD-L1 expression on MDSCs in mice melanoma.

The V-domain Ig suppressor of T-cell activation (VISTA), also designated as PD-1H, is a recently defined negative regulator mediating immune evasion in cancer [119,120], in addition to PD-L1. High VISTA expression levels were found on MDSCs, monocytes, and leukemic blasts, but not on CD4+, CD8+, or T reg cells in acute myelogenous leukemia [121]. The combined inhibition of both the VISTA and PD-1 pathways synergistically improved anti-tumor T-cell activity. Leukemia growth was further diminished by PD-1H blocking antibodies [122,123,124]. Further exploration of PD-1 PD-1H therapy is awaited and in progress.

## 4. Perspectives and Future Directions


(A)Review of Onco-immuno micro-environmental studies in MPN
(1)Inflammatory cytokines and ROS formation.


Chronic inflammation and oxidative stress have been considered to be the hallmarks of MPNs for the last decade [125,126,127]. Inflammatory cytokines have been reported to be increased in murine models and in patients samples, including interleukins IL-1β, IL-6, IL-8, IL-10, IL-11, IL-12, IL-15, IL-17, and IL-33, chemokine (C-X-C motif) ligand 1 (CXCL1), CXCL4, tumor necrosis factor-α (TNF-α), transforming growth factor-β (TGF-β), granulocyte macrophage-colony stimulating factor (GM-CSF), platelet-derived growth factor (PDGF), vascular endothelial growth factor (VEGF), and angiopoietin-1 [128,129,130,131,132,133]. Increased ROS results from chronic inflammation in MPN has also been well described [134,135,136].

Allogeneic stem cell transplantation can lead not only to a complete restoration of hematopoiesis, but also to the regression of bone marrow fibrosis in patients with myelofibrosis. Therefore, neoplastic clones were proposed to be the main driver of this inflammatory reaction, as demonstrated by the curative effect of transplantation [137,138,139]. However, the observation of a gastrointestinal stromal tumor (GIST) following *Helicobacter pylori* infection, enteropathy-associated T cell lymphoma, and adenocarcinomas in patients with coeliac disease supports the idea that inflammation is the initial driver leading to carcinogenesis. Therefore, to argue whether inflammation or abnormal clones are the first initiators of MPN is similar to arguing whether the chicken or the egg came first [140].

The inflammatory cascade was initiated either from (1) the danger associated protein (DAMP), S100A9 and S100A8, which are belong to the subfamily termed myeloid-related proteins that are linked to innate immune functions and are predominantly expressed in the myeloid lineage, including monocytes, neutrophils, and activated macrophages, [141,142], or (2) HMGB1. HMGB1 is an intracellular DNA-binding protein important in chromatin remodeling that can be released by necrotic cells passively and by active secretion from macrophages, natural killer (NK) cells, and dendritic cells and were also associated with inflammation and cancer. (3) PAMPs (pathogen-associated molecular pattern); if the pathogen cannot be eliminated, then it will elicit a chronic inflammatory response [143,144]. These ligand proteins then binds to TLRs and triggers the major pathways, including the MyD88-dependent pathway, which activates NF-kB, as well as through minor pathways involving the mitogen-activated protein kinases (MAPKs) p38 and c-Jun N-terminal kinase (JNK), and then activates interferon regulatory factor 3 (IRF3). Both pathways then form ROS [125,135,145,146]. The continued inflammation results in the accumulation of more ROS in cells, damages DNA, and causes clonal proliferation, driving disease progression towards full-blown MPN [147,148,149].

Therefore, in contrast to the clonal theory for the MPN pathogenesis with Jak-2, mutant clonal activation induces inflammatory changes, the accumulation of more ROS, and more clonal evolution, leading to different phenotypes of MPN [135,150,151,152,153,154]. We have studied the inflammatory pathway in MPN in 97 MPN patients [155], and we found that TLR2 was the predominant pattern recognition receptor (PRR), especially in polycythemia vera (PV) and essential thrombocythemia (ET) in comparison to patients with myelofibrosis (MF). TLR-4 was also elevated, but not as remarkably as TLR-2. ROS production was remarkedly higher in MF than in PV or ET, which implies that the inflammatory process in MPN involves a major role of TLR-2 and a minor role of TLR-4 in accumulating ROS. This leads to DNA damage, and, with years, the accumulation of more ROS formation, DNA damage, and then the transformation of PV or ET into MF. We also found that S100A 9 was increased in MPN, which was also reported by Kovacčicć et al. [156]. S100A9 was found to be associated with colitis-induced colon carcinoma [142] and the formation of MDS in animal models [69,70]. For other molecules, including HMGB1, we only studied a small number of patients, and the molecules were not elevated.



(2)Increased MDSC. In the inflamed micro-environment of MPN, the production of many inflammatory cytokines along with elevated S100A9 results in the accumulation of MDSC in MPN [157]. The mechanism of increased MDSC could be due to (1) the inflammatory cytokine stem cell growth factor (SCF) leading to the accumulation of MDSC [158]; (2) increased S100A9 levels, which inhibited dendric cell maturation and then increased MDSC [70], (3) the cytokine release of GM-CSF, VEGF, PGE2/COX2 (prostaglandin E2/cyclooxygenase-2), and interferon (IFN)-γ. These factors are responsible for MDSC accumulation and C5a, which facilitates MDSC infiltration into tumors and enhances their suppressive abilities [159].(3)Immune dysfunction in MPNs No differences between healthy donors and MPN patients were found in Th cells (T help) polarization at baseline level [160]. The frequency of thymus-derived regulatory T cells (Tregs) has also been studied in MPNs, and conflicting results have been published [161,162,163]. Natural killer (NK) cells in MPN also have a decreased function and numbers [159,164]. Ramano et al. [165] further studied MPN according to the JAK2 and CALR mutation status and reported that patients carrying the JAK2 (V617F) mutation had a reduction in Th17, myeloid-dendric cells (DCs), and effector Tregs, as well as increased ILC1 (hypofunctional lymphocytes) and cytokine-producing Tregs. The CALR-mutated patients revealed high ILC3 levels, reduced Th1, and a reduced capacity of monocytes to mature into fully committed DCs in vitro. Their Tregs were also less effective in inhibiting the proliferation of autologous effector T cells due to an increased proliferative status induced by CALR mutation. Triple-negative patients presented a reduced amount of total circulating CD3, effector Tregs, and Th1, as well as increased ILC1. Keohane et al. [160] reported that CD4+, CD127low, CD25high, and FOXP3+ T regulatory cells are reduced in MPN patients compared to healthy controls. They also reported that this decrease is even more pronounced following JAK2 inhibitor therapy. After 6 months of treatment, the number of T helper (Th)-17 cells increased, and there was a blockade of pro-inflammatory cytokine production, which explained why ruxolitinib therapy increased the incidence of infections. Interestingly, in CALR mutant MPN, a mutant-specific sequence generated by a frameshift mutation has been reported as a neoantigen for CD4+ T cells, but this response was reduced in cells derived from MPN patients harboring a CALR mutation [166,167,168]. Immune checkpoint inhibitor (CPI) therapy enhances shared neoantigen-induced T cell immunity directed against mutated calreticulin in myeloproliferative neoplasms [168].




(B)Review of CPI studies in MPN. There have been very few studies on CPI therapy in MPN, especially in poor prognostic entities such as primary myelofibrosis (PMF), post-ET MF, or post-PV MF. Three NCI-sponsored clinical trials related to CPI therapy (NCT03065400, NCT02421354, and NCT02871323) were listed in 2021(clinicaltrials.gov, accessed on 1 October 2018). NCT02421354 [10], using nivolumab, was terminated due to a low efficacy. NCT02871323 was withdrawn because of a low enrollment, while NCT01822509 assessed ipilimumab in comparison with nivolumab in phase 1 studies. Another two clinical trials were NCT03566446 (Phase I), a CALRLong36 peptide (exon 9 mut) vaccine trial, and NCT04051307, a PD-L1Long [18,19,20,21,22,23,24,25] ArgLong2 [153,154,155,156,157,158,159,160,161,162,163,164,165,166,167,168,169,170,171,172,173,174,175,176,177,178,179,180,181,182,183,184,185,186,187,188] vaccine trial; both studies were vaccine trials based on mutated calreticulin-induced T cell immunity [166,167]. No clinical trials using CPI in the treatment of myelofibrosis or MPN have been listed in 2022 (clinicaltrials.gov, accessed on 1 October 2018).Only one small-scale study of CPI therapy in *Ph*(-) MPN [10] has been reported so far, in which, it was shown to be ineffective, and no clinical trial of CPI in the treatment of MPN has been listed in 2022 among the NIH clinical trials. MDSCs are considered as important to CPI therapy resistance [73,80,169,170,171]. Therefore, the failure of CPI therapy in *Ph*(-) MPN is most likely related to MDSCs.(C)Future Directions. Therefore, from reviewing inflamed microenvironments and deranged immune dysfunction in MPN resulting in the overactive MDSC and being resistant to CPI therapy in solid tumor history, targeting myeloid-derived suppressor cells is a promising strategy to overcome resistance to immune checkpoint inhibitors. Table 1 lists all of the agents that may reduce MDSC numbers, cause them to differentiate into dendric cells, or reduce their PD-L1 levels. The inhibition of VISTA expression levels on MDSCs also represents future work to improve CPI therapy for the treatment of MPN diseases.These listed agents that decrease MDSCs could be added to CPI to improve the CPI therapy efficacy in *Ph*(-) MPN.
(a)Ruxolitinib in combination with CPIIn November 2011, ruxolitinib became the first Janus kinase (JAK) inhibitor approved by the United States FDA to treat myelofibrosis [172]. Recently, ruxolitinib was approved to treat steroid-refractory graft-versus-host disease (GvHD). Many JAK inhibitors have been developed, each with varying activities against other kinases and differential effects on the immune system [173]. In multiple myeloma (MM), PD-L1 expression is increased in plasma cells from patients with MM compared with that from healthy donors, and its expression is associated with a resistance to a variety of anti-MM treatments. The reported inhibition of the PD-L1/PD-1 pathway in plasma cells by ruxolitinib [174] and decreased PMC-MDSC (LOX-1 expression) in Hodgkin’s lymphoma treated with a PD-1 inhibitor plus ruxolitinib [175] suggest that ruxolitinib may be a suitable candidate to be added to CPI therapy in the treatment of MPN, because ruxolitinib is expected to reduce PD-L1 expression in MDSCs. We also found that MDSCs in MPN show an increased PD-L1 expression [176], and ruxolitinib added to the CPI therapy may thus be a promising treatment option.(b)IMID (Immunomodulatory imide drug) therapy in combination with CPIIMID therapy has been reported to have a 30% response rate in treating patients with myelofibrosis, and, in some cases of anemia, patients have changed from transfusion-dependent to transfusion-independent states [177,178]. In a study by Görgün, et al., both newly diagnosed multiple myeloma and relapsed myeloma cells present increased PD-L1 mRNA expression on the myeloma and MDSC cells compared to those from healthy individuals. PD1/PD-L1 blockade abrogated BM-stroma cell (BMSC)-induced MM growth, and lenalidomide decreased PD-L1 expression on myeloma and MDSC cells. The combined blockade of PD1/PD-L1 with lenalidomide further inhibited BMSC-induced tumor growth and decreased PD-L1 on the MDSC and myeloma cells, further improving T cell immunity [179]. Therefore, IMID in combination with CPI may also be worth exploring.(c)BTK inhibitors with CPIBTK inhibitors effectively treat B-cell malignancy, especially chronic lymphocytic leukemia [180]. MDSCs express BTK, and treatment with BTK inhibitors significantly inhibits MDSCs by impairing nitric oxide production and cell migration. In addition, ibrutinib was found to inhibit the in vitro generation of human MDSCs [89] and induce MDSCs to mature to dendric cells in mouse breast cancer models [115]. Reports have also shown a role for BTK in Toll-like receptor (TLR) signaling in myeloid cells [181], which is of interest because TLR signaling has been implicated in MDSC generation and function [182,183]. Therefore, BTK inhibitors in combination with CPI should be considered in MPN.(D)In the future, other newly developed compounds (LCL161, LSD1 inhibitor (bomedemstat), Pelebresib (CPI-0610), AVID200 (TGF-β 1/3 inhibitor) [184,185,186,187] should also be explored in combination with CPI therapy.


## Figures and Tables

**Figure 1 ijms-23-05837-f001:**
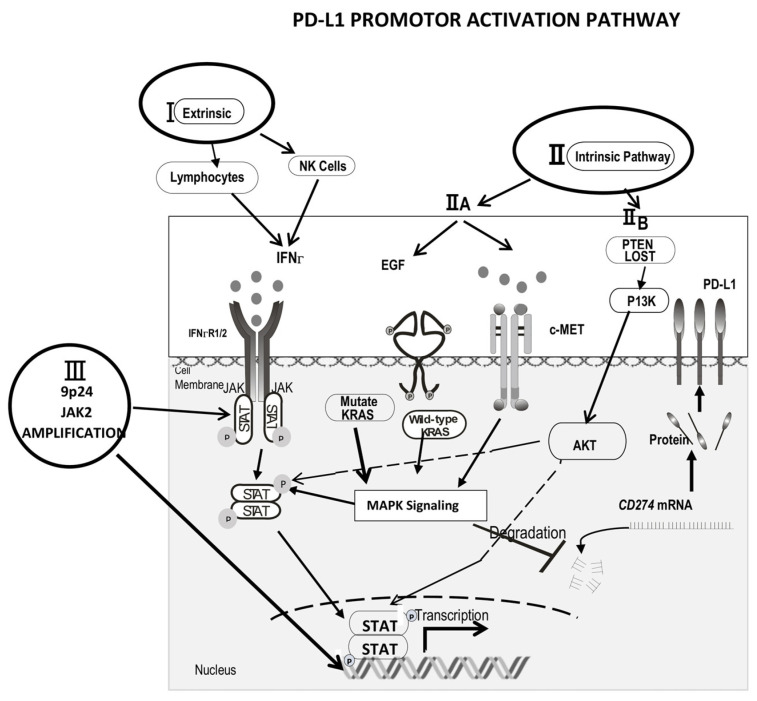
Proposed model for the three major mechanisms of PD-L1 activation. (1) The extrinsic mechanism: tumor-infiltrating lymphocytes (TIL) and NK cells produce IFN-γ. Then, IFN-γ signals through the interferon-gamma–JAK1/JAK2–STAT1/STAT2/STAT3–IRF1 axis, resulting in binding of the IRF1 transcription factor to the PD-L1 promoter (ISRE_s_ and GAS_S_). (2) The intrinsic mechanism: (A) the PD-L1 protein is up-regulated through EGFR/RAS/MAPK pathway activation. Wild-type EGFR activates PD-L1 through the JAK2/STAT1 pathway, whereas mutant EGFR/RAS may induce stronger MAPK pathway activation and STAT3 pathways then activate promotor. Also MAPK signaling increases the stability of *CD274* mRNA. (B) Functional loss of tumor-suppressive genes, such as PTEN, results in activation of PI3Kinase, then activation of the AKT pathway (3) 9p24.1 gene amplicon amplification. The PD-1 ligand genes, PD-L1 and PD-L2, are located on chromosome 9p24.1. A 9p copy gain has been described in Hodgkin’s lymphoma and primary mediastinal large B-cell lymphoma (MLBCL). The amplification region also included the Janus kinase 2 (*JAK2*) locus. This gene amplification is also described in carcinoma of the lung uterine cervical squamous cell carcinoma and triple-negative breast carcinoma. (Permission to modify the original figure (*J. Pathol.*
**2019**, *249*, 52–64) was obtained from Dr. Jong.).

**Table 1 ijms-23-05837-t001:** Anti-MDSC in Cancer.

Anti-MDSC in Cancer			
		Agent	
1	Depleting MDSC	1a. Chemotherapy: 5FU, Gemcitabine 1b. Ibrutinib	2. inhibit VEGF, angioenesis, STAT 3 of tumor microenvironment: sunnitinib etc.
2	Blocking MDSC recruitment	CCR5 inhibitor, CSF1R inhibitor	3. NLRP pathway inhibitor
3	Attenuating the immunosuppressive mechanisms of MDSC	1. COX2 inhibit PGE2 then inhibit arginase	2. Triterpenoid activate Nrf2 to reduce ROS formation
4	Induction of Differentiation of MDSC	1. ATRA induced differentiation of MDSC in both mice and patients in various cancer types, such as renal cell carcinoma	2. 5-azacytidine Docetaxel induce MDSC to TAM, 3. Ibrutinib induced MDSC to DC
5	Decreasing VISTA, PD-L1 expression on the MDSC	PD-1, PD-L1 antibody and anti-Vista antibody	

TAM, tumor associated Macrophage; NLRP, Programmed death ligand 1/NOD-, LRR-, NLRP, pyrin domain–containing protein 3 inflammasome signaling cascade; VISTA, the V-domain Ig suppressor of T-cell activation.

## Data Availability

Not applicable.

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
