# Peer review of "PD-1/PD-L1, MDSC Pathways, and Checkpoint Inhibitor Therapy in Ph(-) Myeloproliferative Neoplasm: A Review"

_ijms, 2022, doi:10.3390/ijms23105837_

Round 1
Reviewer 1 Report
Please see attached.

Author Response
I already had the English edited by IJMS see attached certificate

Reviewer 2 Report
The manuscript submitted for publication reviews the potential of immune checkpoint inhibitor (CPI) treatment in Ph-negative MPNs. CPI is not recognized as an effective treatment option in MPN, so a collection of recent advancements in the form of a thorough review is valuable to the scientific community and would interest the readers.
The authors do provide interesting data regarding PD-1 and PD-L1 expression, and more importantly, regarding myeloid derived suppressor cells (MDSC). The manuscript, however, suffers from several shortcomings.
The text requires extensive editing and spell checking, English usage should be improved. Several passages are very difficult to comprehend because of poor editing.
Some examples of typographical and editing errors:
promotor, MARK pathway, 9P 24 amplicon, Stat 1
The content of the review is suboptimal. The review does not describe Ph-negative MPN or myelofibrosis. The authors do not describe the unmet clinical need for immune checkpoint inhibitor therapy in case of this malignancy.
MDSCs are not presented in detail. It would be interesting to show available literature data regarding the role of MDSCs in the bone marrow microenvironment in case of MPNs. Valuable sources could be cited in this topic. The authors explain the role of IFN-gamma and EGFR/RAS pathways influencing PD-L1 expression, however, they do not show why these are relevant in case of MPNs. Same is true for 9p amplification.
Overall, there is very little about MPNs in the submitted manuscript. The overall poor-quality editing, very few images and lack of sufficient detail regarding the disease targeted by the title reduces the value of the manuscript significantly. Acceptance cannot be recommended.
Author Response
1) We had English edited by IJMS, see the attached certificate.
2) We had changed the word MARK to MAPK , 9P24 to 9p24, Stat1 to stat 1.
3) We had MDSC on MPN data , added to the manuscript ( see page 8, line 25-29) , we also summarized about the clinical trials by NIH in the year of 2021 and none in 2022 of CPI therapy in Myelofibrosis. ( Page 8, line 16-24) .
4) There are no data regarding tumor environment with IFN -gamma and 9p24 in MPN , because it was not studied yet , as in there were many data in solid tumor regarding these Subjects . So in tases subjects , it will need further studies .
5) Our review represents an oncologist also hematologist reviewing about MPN , because many MPN experts only take care MPN as they don't deal with solid tumors.

Reviewer 3 Report
I feel this review article does not reach the level of priority for publication in IJMS and may be more suitable for a specialized journal.
Author Response
1) We had English edited by IJMS , see attached certificate .
2) This review represent an oncologist and hematologist review regarding this subject . PD-1 and MDSC have been extensively studied in solid tumors not in MPN . Because many MPN experts only take care MPN patients but not solid tumors . so extensive review by an oncologist in this subjects will be valuable in MPN .

Round 2
Reviewer 2 Report
Extensive editing of the manuscript was performed, the quality of the text improved significantly.
However, the content-related comments have not been addressed. The manuscript is a review of CPI therapy only barely mentioning MPNs. The authors indicate that CPI therapy is not effective in MPN numerous times in the manuscript and state that information regarding the microenvironment is lacking. The authors only briefly mention some previous studies of their own regarding PD-L1 expression and MDSCs in MPN, although these would be interesting to the readers to better understand the biology of MPN.
This valuable review describes possible mechanisms of resistance to CPI therapy, however, it is not about MPN. It does not introduce the readers to MPN or the unmet clinical need that may indicate the need for CPI. The manuscript does not provide insight into the realm of the inflammatory microenvironment or MDSCs in the case of MPNs. This would be important, since the authors suggest that the bone marrow microenvironment in MPN may provide a unique background of MDSCs that could explain the failure of CPI therapy. This concept, however, is not represented well in this review. Wouldn’t the bone marrow microenvironment similarly effect CPI therapy in case of bone marrow metastasis of a solid tumor? Is there any data indicating limited success in this regard? What about other solid tumors that do not show good response to CPI therapy? Are there similarities in the microenvironment or are MPNs special in this context? No doubt the literature is scarce in this regard, but the authors should provide the readers some data regarding the issue.
If the authors intend to help the hematologist community to have a better understanding of why CPI therapy is not successful in MPN and provide insight, then the main text should be modified accordingly. Otherwise, the title of the manuscript should be changed (one suggestion: PD-1/PD-L1 and MDSC pathways in immune checkpoint inhibitor therapy: reasons of failure) and then MPNs are just mentioned as examples.
Author Response
Now we have reviewed the on-immuno-microenvironments in MPN on Page 10 , line 1 through Page 11 line 10. and we stated that with this inflamed micro-environment resulting overactive MDSC , likely were related to the CPI failures in MPN.
Reviewer 3 Report
The title of the review does not reflect the content in the body of the manuscript. The authors need to focus more on the Ph negative myeloproliferative neoplasm as not much has been discussed by authors in this review about the checkpoint therapy or pathways for the disease
Author Response
We had added From Page 10 line 1 to Page 11 , line 10 about the review of inflamed micro-environments in MPN
Round 3
Reviewer 2 Report
The authors responded to previous concerns by adding approximately a page of additional information regarding the inflammatory background of MPN.
Extensive editing by a proficient English speaker is again necessary before acceptance may be considered, otherwise there are no further issues raised.
Author Response
I had sent my manuscript to IJMS English service (See Attached files B)

Reviewer 3 Report
Not sure what changes authors have made to the manuscript.
Author Response
I had sent the manuscript To IJMS English Edition Service for English edition
